

# Understanding perceived exertion in power-aimed resistance training: the relationship between perceived exertion and muscle fatigue

Hanye Zhao[1], Takanori Kurokawa[2], Masayoshi Tajima[2], Zijian Liu[2] and Junichi Okada[1]

[1] Faculty of Sport Sciences, Waseda University, Tokorozawa, Saitama, Japan
[2] Graduate School of Sport Sciences, Waseda University, Tokorozawa, Saitama, Japan

## ABSTRACT

**Background**. Perceived exertion is an inevitable outcome of power-oriented resistance training. However, it remains unknown whether perceived exertion is dominated by central or peripheral responses during this type of training. This study examined the effect of neuromuscular responses on the determination of ratings of perceived exertion (RPE) during power bench press (BPress) exercise.

**Methods**. Sixteen trained males performed three BPress tasks with varying volumes (low, medium, high) at 65% of their one-repetition maximum. RPE, surface electromyography, and velocity loss were assessed across all conditions. Peak root mean square (RMS) and median frequency (MDF) were calculated from the surface electromyography data.

**Results**. Significant effects were observed across experimental conditions for overall RPE, average velocity loss, and average MDF (all $p < 0.001$), while no significant difference was found in average RMS. As the lifting tasks progressed, significant effects of repetition were observed in all measured variables (all $p < 0.001$). When comparing conditions, significant differences were found among the three in RPE, velocity loss, and MDF (all $p < 0.001$), whereas no significant effect of condition was observed in RMS. No significant relationship was found between MDF and velocity loss.

**Conclusions**. In power BPress, higher repetitions affected RPE, velocity loss, and MDF, while peak RMS was less responsive. These findings indicate that both athletic performance and RPE are primarily influenced by peripheral fatigue. However, velocity loss should be interpreted with caution as a fatigue marker in this context.

# INTRODUCTION

Strong evidence suggests that resistance training is effective in maintaining and enhancing muscle mass, functional capacity, and athletic performance (*Garber et al., 2011*). However, when performing resistance exercises, individuals often experience subjective sensations such as fatigue and exertion (*Ekkekakis, Parfitt & Petruzzello, 2011*). A previous study reported that perceived exertion is associated with endurance performance outcomes, such

Corresponding author
Hanye Zhao,
zhaohanye@toki.waseda.jp

as time to exhaustion (*Di Fronso et al., 2020*). Therefore, understanding these subjective exertion is crucial for developing appropriate resistance training guidelines, achieving optimal athletic performance, and designing long-term periodized training programs.

Perceived exertion during physical exercise can be quantified using the rating of perceived exertion (RPE) scale. RPE is a subjective, perception-based method commonly used to assess exercise intensity (*Lagally et al., 2002a*; *Kilpatrick et al., 2009*; *Lea et al., 2022*). Originally developed for evaluating the intensity of rehabilitation and other medical treatments, the RPE scale has since been widely adopted in physiological and sports-science research due to its simplicity and its validity in reflecting physiological responses to physical activity (*Robertson et al., 2004*; *Fontes et al., 2010*). Over the past 30 years, with the growing popularity of resistance training worldwide, researchers and practitioners have increasingly incorporated RPE into resistance exercise protocols as supporting evidence for exercise prescription (*Lagally & Robertson, 2006*; *Gearhart et al., 2009*; *Zhao, Nishioka & Okada, 2022*). During aerobic exercise, many factors affect RPE, including environmental temperature, body temperature, and hypoxia (*Tucker, 2009*). In resistance exercise, several key factors have been examined as contributors to increased RPE, including load, volume, and rest-interval configuration (*Lagally et al., 2002a*; *Hiscock et al., 2018*; *Zhao, Yamaguchi & Okada, 2020*). As a result, RPE has become a valuable physiological marker for resistance training, offering both practicality and insight.

Increased fatigue levels have been identified as one of the primary contributors to elevated RPE scores (*Fontes et al., 2010*; *Blain et al., 2016*; *Weakley et al., 2017*; *Cruz-Montecinos et al., 2019*). However, most previous studies have examined the relationship between RPE and either peripheral (*e.g.*, cortisol, blood lactate) or central (*e.g.*, command from motor cortex) responses in isolation, typically under controlled laboratory conditions (*De Morree, Klein & Marcora, 2012*; *Vargas-Molina et al., 2020*). As a result, it remains unclear whether RPE during actual resistance exercise scenarios is predominantly influenced by central or peripheral factors.

With the growing adoption of velocity-based training, velocity loss has been recognized as an indirect marker of fatigue in resistance exercise (*Sánchez-Medina & González-Badillo, 2011*). Studies have demonstrated that velocity loss is significantly correlated with fatigue-related metabolic markers, such as blood lactate and ammonia concentrations, during power-aimed resistance exercises (*Izquierdo et al., 2006*; *Sánchez-Medina & González-Badillo, 2011*). However, despite its growing application, the validity of velocity loss as a fatigue marker lacks sufficient neuromuscular evidence.

Surface electromyography (sEMG) is a non-invasive tool widely used to assess neuromuscular function in physiological, sports science, and clinical settings (*González-Izal et al., 2012*; *Campanini et al., 2020*). When sEMG is used to grasp fatigue induced by peripheral mechanisms, changes in power spectral density serve as a marker of fatigue (*Brody et al., 1991*). These changes, caused by intramuscular metabolic disturbances, result in a shift of the power spectrum toward lower frequencies (*Dimitrov et al., 2006*; *González-Izal et al., 2012*; *Campanini et al., 2020*). In contrast, when sEMG is used to measure central drive, signal amplitude is typically employed as an indicator of motoneuronal activity (*De Morree, Klein & Marcora, 2012*; *Blain et al., 2016*; *Campanini et al., 2020*). Overall, sEMG

provides a valuable means to investigate both peripheral and central nervous system responses during muscle contractions.

Consequently, the use of RPE to quantify exertion during resistance training has increased; however, it remains unclear whether exertion is influenced predominantly by central or peripheral factors. Accordingly, elucidating the contributors to RPE using neuromuscular evidence may help practitioners in training, sports science, and clinical settings better understand the relationship between perceived exertion and actual performance. The present study aims to: (1) examine the relationship between RPE and neuromuscular fatigue, as assessed by sEMG, to determine whether RPE is primarily influenced by central or peripheral responses; and (2) evaluate the validity of velocity loss as a fatigue indicator by analyzing its relationship with neuromuscular fatigue during power bench press (BPress) exercises. We hypothesized that (1) peak root mean square (RMS) would increase and (2) median frequency (MDF) would decrease as RPE increased. However, (3) velocity loss may reflect neuromuscular fatigue only under certain conditions.

## MATERIALS & METHODS

### Experimental designs

This study aimed to examine the relationship between RPE and neuromuscular fatigue during power BPress exercise. Additionally, it sought to evaluate the validity of velocity loss as an indicator of neuromuscular fatigue during power BPress. A counterbalanced crossover design was employed. Power BPress was chosen as the experimental exercise due to its widespread use in resistance training and velocity-based training protocols (*Sánchez-Medina & González-Badillo, 2011*). The protocol consisted of two experimental sessions, separated by at least 48 h. In the initial session, participants' descriptive information was collected, and they were introduced to the Borg CR-10 scale. An anchoring trial was then conducted to establish the upper and lower limits of RPE for use in the subsequent session (*Lagally et al., 2002b*; *Zhao, Seo & Okada, 2024*). This trial involved performing a single set of BPress to volitional failure. The number of successfully completed repetitions was recorded to determine the repetition volume for the following experimental conditions. In the second session, participants performed 30% (low; L), 60% (medium; M), and 90% (high; H) of the repetitions completed during the anchoring trial (*Zhao et al., 2024*). The L, M, and H conditions were administered in a counterbalanced order. For each condition, sEMG signals, RPE scores, and velocity data were recorded.

### Participants

The required sample size was determined using G*Power 3.1 (University of Bonn, Bonn, Germany) based on an ANOVA with fixed effects, special, main effects and interactions. The input parameters were set as follows: effect size = 0.4, alpha = 0.05, and statistical power = 0.95 (*Cruz-Montecinos et al., 2019*). The present study included three conditions, and three repetitions were extracted for the ANOVA; therefore, the numerator degrees of freedom were four and the number of groups was nine. The analysis indicated a minimum required sample size of 14 participants. Accordingly, 16 male participants were recruited for this study. All participants were male college students with regularly

engaged in resistance training (training experience: 4.6 ± 2.3 years). None reported any history of neuromuscular or musculoskeletal injuries, nor were they undergoing medical treatment at the time of participation. The descriptive characteristics of the participants (mean ± SD) were as follows: age = 20.8 ± 1.5 years, body mass = 72.9 ± 12.2 kg, height = 171.8 ± 5.6 cm, body fat percentage = 15.9 ± 4.0%, and one-repetition maximum (1RM) for the bench press = 93.6 ± 17.3 kg. All participants were fully informed of the experimental procedures, including the measurement items, potential risks, discomforts, and benefits. Written informed consent was obtained prior to participation. The study was conducted in accordance with the Declaration of Helsinki and approved by the Human Ethics Committee of Waseda University (Approval No. 2023-112).

### Initial session

Before the measurements began, participants received detailed instructions regarding the purpose of the study, the measurement items, and the potential risks and benefits. Written informed consent was then obtained. Subsequently, participants' descriptive characteristics were recorded, including body fat percentage, which was assessed using a bioelectrical impedance device (InBody 970; InBody Co., Ltd., South Korea). Following these assessments, participants completed a standardized warm-up protocol consisting of five minutes of jogging, static and dynamic stretching, and two sets of power BPress using 20 kg and 30 kg for six and eight repetitions, respectively. This warm-up procedure has been reported to enhance explosive performance (*Turki et al., 2011*). After the warm-up, participants' 1RM for the bench press was assessed following the guidelines of the National Strength and Conditioning Association (*Baechle, Earle, and National Strength & Conditioning Association, 2008*). The 1RM test involved progressively increasing the load until the participant successfully completed one repetition at their maximum capacity. Exercise technique was checked and monitored by a certified trainer. The range of motion and criterion for a successful repetition were defined according to a previous study to prevent rebound at the bottom position and to minimize artifacts in the sEMG signals (*Zhao, Seo & Okada, 2024*).

After recording descriptive characteristics and completing the 1RM test, participants rested for five minutes. During this interval, they received detailed instruction on the Borg CR-10 scale, which ranges from "nothing at all" (0) to "extremely strong" (10), with verbal descriptors for perceived exertion (*Borg, 1998*). In this study, RPE was defined as the subjective sensation of exertion experienced in the upper body during the BPress exercise. Following the RPE scale instruction, participants were introduced to the prescribed lifting cadence. To minimize the effect of barbell rebound during the BPress, the eccentric (lowering) phase was set at a 2-second duration, followed by an explosive concentric (pushing) phase, in accordance with previous protocols (*Zhao et al., 2025*). The cadence was regulated using a smartphone metronome emitting one beep per second. A safety bar was positioned just above chest level to prevent unintentional rebound and to avoid impact on sEMG signals in the subsequent experimental session. After the rest period, participants completed a practice set using only the barbell, following the prescribed cadence and technique. Then, an anchoring trial was conducted to establish the subjective

exertion range for use in the experimental session. Participants performed a single set of power BPress to volitional failure. The low anchor of RPE (score of 0, "nothing at all") was defined as the sensation of resting in a relax state (*Eston, James & Evans, 2009*), while the high anchor (score of 10, "extremely strong") was defined as the feeling experienced at the point of failure (*Mayo, Iglesias-Soler & Kingsley, 2019*). The relative intensity was set at 65% of each participant's 1RM, consistent with the subsequent experimental conditions (*Robertson et al., 2003*). A repetition was considered successful if the participant lowered the barbell to the safety bar and then returned it to the starting position with full control.

## Experimental session

Before the start of the experimental session, participants completed the same standardized warm-up routine as in the initial session, which included jogging, static and dynamic stretching, and two sets of power BPress. Following the warm-up, participants engaged in an RPE reporting practice session. To facilitate RPE reporting, the Borg CR-10 scale was positioned vertically above the participants' heads for easy visibility. The lifting cadence was set at 2 s for the eccentric phase, followed by an explosive concentric phase, with a 2-second pause between repetitions. This cadence was controlled using a metronome emitting one beep per second. To capture intra-set RPE without disrupting technique, a 2-s pause was imposed between repetitions. During each pause, participants verbally reported the RPE for the latest repetition using the Borg CR-10 scale (*Zhao et al., 2025*). They were instructed to base their responses on the subjective range established during the anchoring trial. If participants experienced a sensation of exertion exceeding the high anchor of 10 ("extremely strong"), they were permitted to report values greater than 10, in accordance with previous recommendations (*Borg, 1998*; *Hollander et al., 2003*).

Following the warm-up and RPE reporting practice, the L, M, and H volume conditions were performed in a counterbalanced order. The required number of repetitions for each condition was calculated based on the number of repetitions completed during the anchoring trial. Previous studies have shown that RPE can be influenced by anticipatory responses to a given task (*St Clair & Timothy, 2004*; *Noakes, St & Gibson, 2005*). To minimize the influence of this central governor effect, participants were not informed of the number of repetitions required for each condition until the second-to-last repetition. They were then instructed to complete one final repetition before stopping. During the 2-second pause between repetitions, participants reported their RPE by selecting a number on the Borg CR-10 scale. As muscle fatigue accumulated during the sets, participants occasionally had difficulty maintaining the prescribed cadence for the eccentric and pause phases. In such cases, they were encouraged to follow the metronome cues as closely as possible. Upon completion of each condition, participants were asked to report an overall RPE for the entire set. A 5-minute rest interval was provided between each condition. The lifting and rest-interval configuration was based on previous studies focusing on fatigue and RPE. Given the participants' relatively greater training experience in the present study, this rest-interval configuration was sufficient for present study design (*Sánchez-Medina & González-Badillo, 2011*; *Zhao et al., 2025*).

## Surface electromyography

sEMG signals were recorded from the dominant side of the pectoralis major, lateral head of the triceps brachii, and anterior deltoid muscles. Bipolar Ag/AgCl surface electrodes (ADMEDEC Co., Ltd., Tokyo, Japan) were used for data collection. For each muscle, two electrodes were placed one cm apart. Prior to electrode placement, the skin was shaved, abraded with sandpaper, and cleaned with alcohol swabs to minimize impedance. Electrode placement followed the standard guidelines for sEMG (*Barbero, Merletti & Rainoldi, 2012*). sEMG signals were recorded using a wireless acquisition system (MARQ MQ-8; Kissei-Com Tech, Nagano, Japan) with a sampling frequency of 1,000 Hz. Data were transmitted in real time to a dedicated computer. A high-speed camera (Grasshopper GRAS-03K2C, FLIR Systems Inc., Canada) was positioned in the sagittal plane to record motion video synchronized with the sEMG signals throughout all lifting tasks. Raw sEMG data were collected using proprietary software (Vital-Recorder; Kissei-Com Tech, Nagano, Japan). Each signal was segmented into individual repetitions based on the synchronized video recordings. Only the concentric (pushing) phases of each repetition were extracted and used for neuromuscular fatigue assessment.

A fourth-order Butterworth bandpass filter (20–450 Hz) was applied to the raw sEMG signals to eliminate background noise. The filtered signals were then used to quantify neuromuscular responses under each experimental condition. Root mean square (RMS) and median frequency (MDF) values were calculated as indicators of neuromuscular fatigue. To enable comparison across repetitions, the peak RMS and MDF values were normalized to the first repetition of each condition, following recommended practices for sEMG normalization during resistance training (*Lanza et al., 2023*). Subsequently, the RMS and MDF values from the pectoralis major, lateral head of the triceps brachii, and anterior deltoid muscles were averaged to produce a single variable for each repetition, which was then used in statistical analyses. Muscle fatigue was identified based on a significant reduction in MDF, while RMS values were interpreted as indicators of changes in central motor output. All RMS and MDF calculations were performed using custom scripts in MATLAB (version 2024a; The MathWorks, Natick, MA, USA).

## Velocity loss

Barbell velocity during the bench press was calculated using a high-speed camera positioned five m from the bench, capturing footage from the sagittal plane on the participant's left side at 120 frames per second. A reflective marker was attached to the end of the barbell to track its vertical trajectory. To convert pixel coordinates to real-world distances, a pixel-to-distance calibration was performed using a 1-meter calibration stick placed approximately at the level of the barbell's path (*Ang & Kong, 2023*). The vertical displacement and concentric phase velocity of the barbell were computed frame-by-frame using dedicated motion analysis software (KineAnalyzer, Kissei-Com Tech, Nagano, Japan). The velocity data were separated into single repetitions and exported for subsequent statistical analysis. Peak value of concentric velocity was extracted using customized MATLAB code (MATLAB 2024a; The MathWorks, Natick, MA, USA). Velocity loss across repetitions was calculated as the percent decrease from the highest to the lowest value of each condition (*Sánchez-Medina*

*& González-Badillo, 2011*). Linear position transducers (*e.g.*, GymAware) were not used because most linear encoders could not be synchronized with the sEMG system and/or lacked sufficient technical transparency (*e.g.*, sampling frequency, coordinate definitions, access to raw time-series data). Instead, barbell velocity was measured with a high-speed camera (120 fps) to enable frame-level synchronization with sEMG.

## Statistical analysis

The Shapiro-Wilk test was used to assess the normality of overall RPE, average velocity loss, RMS, and MDF for each condition. Both average MDF and average velocity loss were normally distributed; therefore, a one-way ANOVA was conducted on these variables, followed by Bonferroni-corrected post hoc tests for multiple comparisons. The overall RPE values for the M and H conditions did not meet the assumption of normality. As a result, Friedman's test was employed to assess differences in overall RPE across the three experimental conditions, with Bonferroni correction applied for post hoc comparisons.

To further investigate changes in RPE, MDF, RMS, and velocity loss during the lifting tasks, data from specific repetitions were analyzed. Due to variability in the number of repetitions performed in each condition (L: $3.4 \pm 1.1$; M: $6.8 \pm 2.2$; H: $10.2 \pm 3.3$), the first, median, and last repetitions were extracted for analysis. A two-way ANOVA (3 conditions × 3 repetitions) was conducted to test for main and interaction effects of condition and repetition (*Zhao, Nishioka & Okada, 2022*; *Zhao, Seo & Okada, 2023*). When significant effects were observed, Bonferroni-corrected post hoc tests were performed.

For correlation analysis, Spearman's Rho coefficients were calculated to assess the relationships between average velocity loss and average MDF. All statistical analyses were performed using SPSS version 29.0 (IBM Corp., Armonk, NY, USA). The level of statistical significance was set at $p < 0.05$.

## RESULTS

Significant differences in overall RPE were observed across the experimental conditions ($\chi^2 = 31.524$, $p < 0.001$). Post hoc comparisons revealed significant differences between L and M ($p = 0.024$), M and H ($p = 0.011$), and L and H ($p < 0.001$) conditions (Fig. 1A). For average velocity loss, significant differences were found between L and M ($p = 0.001$, Cohen's $d = 1.616$), as well as between L and H ($p < 0.001$, Cohen's $d = 2.090$) (Fig. 1B). Significant differences in average MDF were observed between M and H ($p = 0.027$, Cohen's $d = 0.831$) and between L and H ($p = 0.003$, Cohen's $d = 1.288$) (Fig. 1D). In contrast, no significant differences were found in average RMS across the conditions (Fig. 1C).

Intra-set data were analyzed using a two-way ANOVA, with the first, mid-point, and last repetitions extracted for comparison. Significant interaction effects between condition and repetition were observed for RPE ($p < 0.001$, $F = 39.060$, partial $\eta^2 = 0.723$), velocity loss ($p < 0.001$, $F = 13.266$, partial $\eta^2 = 0.469$), and MDF ($p < 0.001$, $F = 19.934$, partial $\eta^2 = 0.571$). A significant main effect of repetition was also found for RPE ($p < 0.001$, $F = 73.833$, partial $\eta^2 = 0.831$), velocity loss ($p < 0.001$, $F = 62.850$, partial $\eta^2 = 0.807$), and MDF ($p < 0.001$, $F = 74.986$, partial $\eta^2 = 0.833$), indicating consistent changes

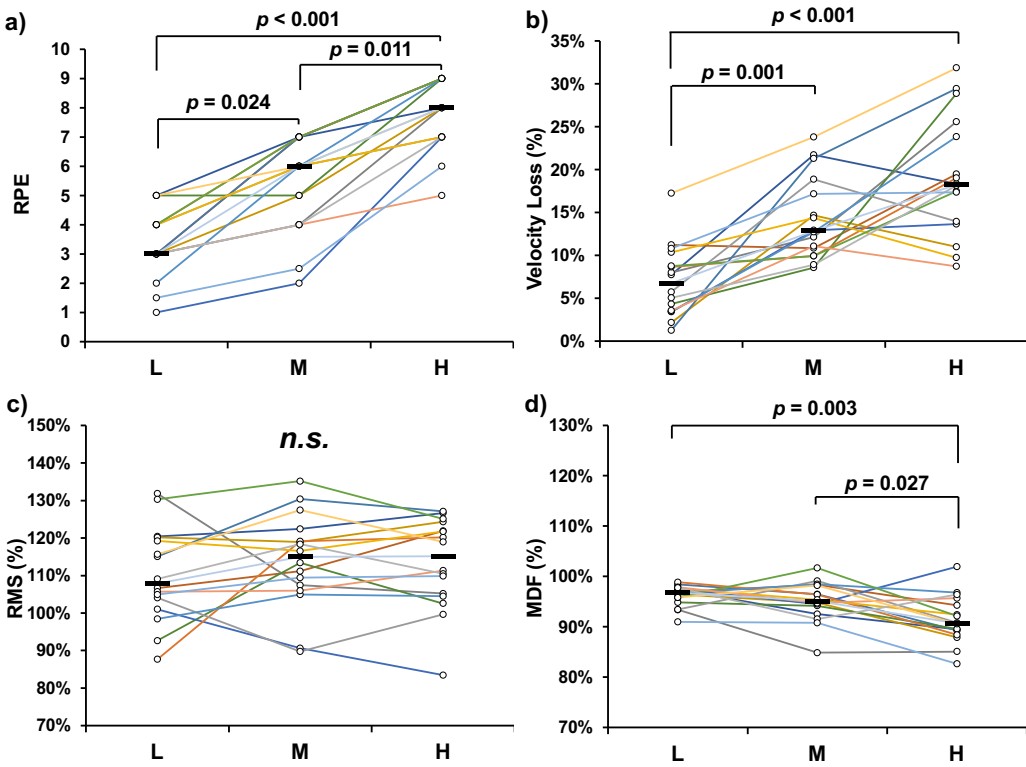

**Figure 1** **Overall ratings of perceived exertion (RPE) (A), average velocity loss (B), average root mean square (RMS) (C), and average median frequency (MDF) (D) of bench press tasks.** The *p* value represents the diûerences and significant level between experimental conditions.

throughout the lifting tasks. Within-condition comparisons showed significant increases in RPE across repetitions for all three conditions (L, M, H; all $p < 0.001$). For velocity loss, significant increases were observed in the M ($p < 0.001$, $F = 32.041$, partial $\eta^2 = 0.681$) and H ($p < 0.001$, $F = 46.172$, partial $\eta^2 = 0.755$) conditions, but not in the L condition ($p = 0.545$, $F = 0.619$, partial $\eta^2 = 0.040$). For MDF, significant decreases were observed in the L ($p < 0.001$, $F = 25.893$, partial $\eta^2 = 0.633$), M ($p < 0.001$, $F = 28.440$, partial $\eta^2 = 0.655$), and H ($p < 0.001$, $F = 77.326$, partial $\eta^2 = 0.838$) conditions. Between-condition comparisons revealed significant differences in RPE at the mid-point ($p < 0.001$, $F = 34.748$, partial $\eta^2 = 0.698$) and last repetition ($p < 0.001$, $F = 73.708$, partial $\eta^2 = 0.831$). Velocity loss also differed significantly between conditions at the mid-point ($p < 0.001$, $F = 9.805$, partial $\eta^2 = 0.395$) and last repetition ($p < 0.001$, $F = 37.089$, partial $\eta^2 = 0.712$).

For MDF, a significant between-condition difference was found at the last repetition ($p < 0.001$, $F = 28.436$, partial $\eta^2 = 0.655$). For peak RMS, only a significant main effect of repetition was found ($p < 0.001$, $F = 22.654$, partial $\eta^2 = 0.602$). No significant main effect of condition or interaction effect was observed for intra-set RMS.

The results of the Spearman correlation analysis are presented in Table 1. No significant correlations were found between MDF and velocity loss across the three experimental conditions.

**Table 1  Spearman's Rho of median frequency-velocity loss during experimental bench press tasks.**
MDF, median frequency; L, low-repetition condition; M, median-repetition condition; H, high-volume condition.

| Pairs | MDF (L)-velocity loss (L) | MDF (M)-velocity loss (M) | MDF (H)-velocity loss (H) |
|---|---|---|---|
| Correlation coefficients ($p$-value) | $-0.312$ ($p = 0.240$) | $0.076$ ($p = 0.778$) | $-0.241$ ($p = 0.368$) |

# DISCUSSION

The purpose of this study was twofold: (1) to examine the relationship between RPE and neuromuscular responses, as indexed by RMS and MDF of sEMG, and (2) to evaluate the validity of velocity loss as an indicator of fatigue during power BPress exercises. The key findings of the study are as follows: (1) MDF, velocity loss, and RPE showed similar trends across conditions, suggesting that the experimental tasks induced fatigue, and that RPE is primarily influenced by peripheral fatigue-related responses. (2) No significant differences were observed between conditions in either intra-set peak RMS or average RMS, indicating that central motor command remained relatively stable despite increasing fatigue levels during power BPress exercises. (3) Although velocity loss increased alongside MDF, no significant correlations were found between the two, suggesting that while velocity loss may reflect aspects of muscle fatigue, it only has limited precision in the context of power BPress.

Subjective exertion during physical exercise is influenced by a range of physiological responses, spanning from central nervous system activity to peripheral muscle fatigue (*Hollander et al., 2003*; *De Morree, Klein & Marcora, 2012*). Previous studies have reported a direct association between RPE and central motor command, demonstrated through various neuromuscular measures during resistance exercise (*Pincivero et al., 1999*; *De Morree, Klein & Marcora, 2012*). For example, concurrent increases in RMS, RPE, and electroencephalogram amplitude were observed during elbow flexion tasks (*De Morree, Klein & Marcora, 2012*). Regarding peripheral mechanisms, RPE has also been shown to correlate significantly with metabolic and endocrine markers such as blood lactate, testosterone, and cortisol concentrations (*Weakley et al., 2017*; *Hiscock et al., 2018*). In the present study, we observed significant increases in RPE accompanied by significant decreases in MDF across all experimental comparisons. In contrast, no significant changes in RMS corresponding to RPE were found. Notably, the trends in average MDF and overall RPE were closely aligned (Figs. 1A and 1D), and RPE increased significantly as MDF decreased within conditions (Figs. 2A and 2D). Although the MDF difference between conditions at the mid-point did not reach statistical significance, the effect approached the threshold ($p = 0.055$), suggesting a possible trend. A decrease in MDF, as measured by sEMG, is typically attributed to a reduction in muscle fiber conduction velocity, which is strongly influenced by local physiological changes such as pH reduction and accumulation of metabolic byproducts. These factors alter the metabolic milieu at the muscular level (*Brody et al., 1991*; *Amann et al., 2011*; *Gorostiaga et al., 2012*; *Keller et al., 2019*). Additionally, these peripheral changes increase afferent feedback from group

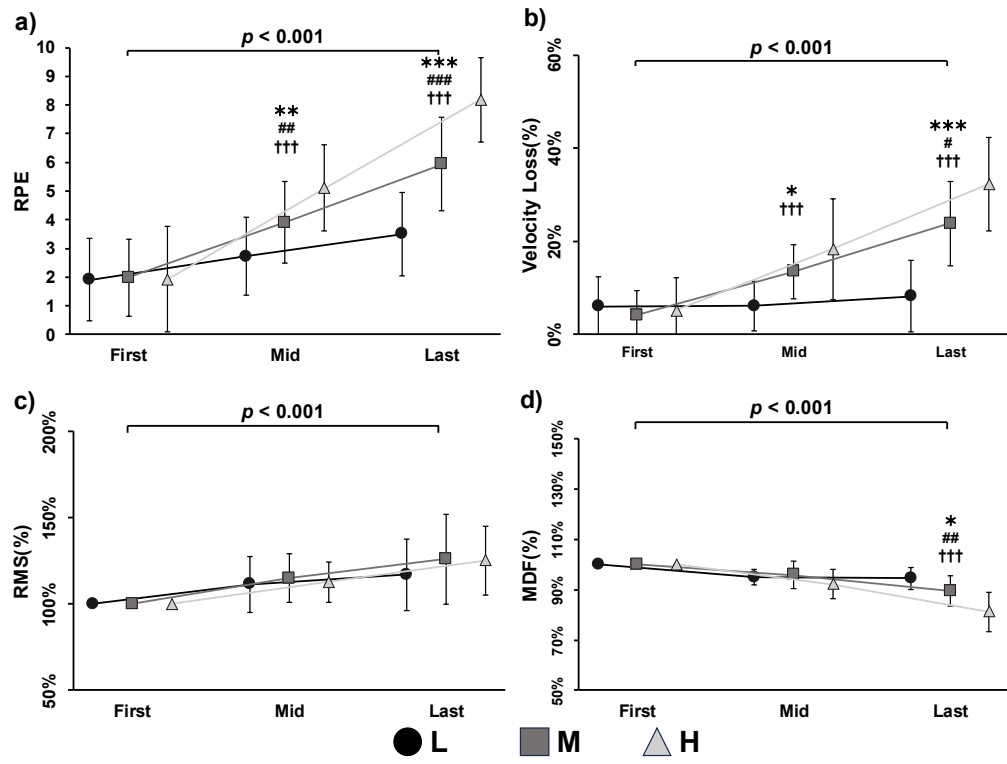

**Figure 2** **Rating of perceived exertion (RPE) (A), velocity loss (B), root mean square (RMS) (C), and median frequency (MDF) (D) during low (L, circle with solid lines), medium (M, square with solid lines), and high (H, triangle with solid lines) volume condition of exercises.** Note: *Represents a significant difference as L compared with M condition, * $p < 0.05$, ** $p < 0.01$, *** $p < 0.001$; # represents a significant difference as M compared with H condition, # $p < 0.05$, ## $p < 0.01$, ### $p < 0.001$; † represents a significant difference as L compared with H condition, † † † $p < 0.001$; the $p$ value indicates the overall main effect and significant level of repetitions.

III/IV muscle fibers, contributing to heightened sensations of exertion (*Blain et al., 2016*; *Broxterman et al., 2017*). Neuromuscular fatigue is typically characterized by a decrease in MDF and an increase in RMS (*Enoka & Duchateau, 2008*; *Cruz-Montecinos et al., 2019*). However, in the current study, RMS did not increase significantly across repetitions in any condition. These findings suggest that fatigue was predominantly localized to peripheral musculature, and that peripheral mechanisms play a key role in RPE determination. In contrast, central motor command—as indicated by RMS—appears to remain relatively stable and may have only a limited influence on subjective exertion during power-aimed BPress exercises.

Previous studies have investigated the mechanisms of fatigue in the context of pedaling exercises, identifying contributions from both central and peripheral factors (*Amann, 2011*; *Amann et al., 2011*; *Blain et al., 2016*). However, these findings may not be directly applicable to resistance training due to the unique characteristics of this modality, such as abrupt intensity shifts driven by rest intervals, the lifting-to-rest phase ratio, and distinct physiological demands (*Paulo et al., 2012*; *Weakley et al., 2017*; *Zhao, Yamaguchi & Okada,*

*2020*). As a result, the specific contributions of central and peripheral fatigue in resistance exercise remain less clearly defined. In the present study, neuromuscular fatigue was assessed indirectly using sEMG, based on the relationship between muscle fatigue and bioelectrical responses. Although peripheral biomarkers (*e.g.*, blood lactate) were not measured, a significant increase in velocity loss was observed alongside a decrease in MDF (Figs. 1B and 1D; Figs. 2B and 2D). These corresponding changes strongly suggest the presence of substantial fatigue induced by the lifting tasks (*Sánchez-Medina & González-Badillo, 2011*). However, no significant increases in RMS were observed, indicating that central motor command remained stable across conditions. This pattern supports the conclusion that, during power-aimed resistance exercises, neuromuscular fatigue is primarily driven by peripheral mechanisms. These findings have important implications for training prescription. For instance, if the goal is to minimize fatigue while maintaining high power output, coaches and personal trainers should avoid scheduling power-aimed training sessions immediately after endurance-focused workouts or game days, as these activities typically involve high workloads and may leave behind residual metabolic byproducts that impair subsequent performance (*Thorlund et al., 2008*). Conversely, the stable RMS levels observed across conditions offer a novel insight for time-efficient training strategies. Since power-aimed training does not appear to demand high central output, it may be effectively combined with strength-oriented sessions involving heavy loads and low repetitions (*Rodriguez-Lopez et al., 2022*). This strategy could help maximize training outcomes without imposing excessive fatigue.

For practitioners such as coaches and personal trainers implementing velocity-based training, accurately assessing fatigue is essential—particularly when the training objective is to maximize power output (*Sánchez-Medina & González-Badillo, 2011*; *Mayo, Iglesias-Soler & Kingsley, 2019*). To make fatigue more observable in real time, velocity loss has been proposed as a reliable indirect marker of neuromuscular fatigue during power-aimed training, and various commercial tools are currently available to facilitate this approach (*Sánchez-Medina & González-Badillo, 2011*; *Martínez-Cava et al., 2020*; *Külkamp et al., 2024*). Most previous studies have validated velocity loss using devices commonly employed in training environments, such as linear position transducers. In contrast, the present study used a high-speed camera and a reflective marker to measure barbell velocity, offering higher temporal precision than many commercial encoders (*e.g.*, 100 Hz). Although linear position transducers have become popular in recent years, there is still a lack of validation against gold standards such as motion capture (*Pérez-Castilla et al., 2019*). Thus, when using velocity loss as a muscle-fatigue indicator, precision may be negatively affected if a linear encoder is used for this purpose. Under these conditions, our findings suggest that velocity loss may not be as reliable as previously reported, and caution is warranted when relying solely on this metric during power-aimed training. Although velocity loss showed a trend similar to MDF, no significant correlations were observed between the two in any condition. Notably, in the L condition, velocity loss did not change significantly during the lifting task ($p = 0.545$; Fig. 2B), whereas MDF exhibited a significant decline ($p < 0.001$). This discrepancy highlights a key limitation: while velocity loss may increase with fatigue, it does not always provide sufficient precision to accurately reflect neuromuscular fatigue.

Moreover, if velocity-based training relies on measurement devices with limited precision, this may lead to underestimation of fatigue levels, potentially increasing injury risk and compromising performance. Therefore, it is recommended that velocity loss be used in conjunction with other fatigue markers—such as jump height and RPE—to enhance the accuracy of fatigue monitoring (*Ilias, 1998*; *Zhao, Seo & Okada, 2024*).

This study has several limitations that should be acknowledged. First, only upper-body power BPress was examined. Prior research has demonstrated that upper- and lower-body exercises may elicit different fatigue responses due to differences in physiological characteristics, such as lactate kinetics (*Van Hall et al., 2003*). Additionally, fatigue responses in the lower body have been reported to be more complex and variable (*Zhao, Nishioka & Okada, 2022*). Second, to capture neuromuscular responses of actual power-aimed training, we used sEMG in present study. While RMS is interpreted as an indicator of central motor command, it is an indirect measure and may not fully capture the complexity of central fatigue. Thus, interpretations regarding central output should be made with caution. Third, although the sample size was determined using statistical power analysis, it remains relatively small, and the predictive model may not be fully suited to the present design (*e.g.*, within- or between-factors ANOVA). For example, the pairwise comparison of MDF at the mid-point approached but did not reach statistical significance ($p = 0.055$), suggesting that a larger sample may be needed to confirm certain effects. Future studies should aim to develop more effective methods for collecting intra-set RPE data, directly compare fatigue responses between upper- and lower-body power-aimed training, and recruit larger samples to enhance the statistical power and generalizability of the findings.

## CONCLUSIONS

The present study demonstrated that MDF, RPE, and velocity loss exhibited similar trends across experimental conditions, while no significant changes were observed in RMS. These findings suggest that fatigue during power-aimed BPress is primarily induced by peripheral mechanisms, and that subjective exertion is largely influenced by peripheral rather than central responses. In contrast, power BPress at moderate intensity does not appear to require substantial central motor command. From a practical standpoint, coaches and practitioners may consider integrating power-aimed training with strength-focused sessions, particularly when time constraints prevent separate implementation of both modalities, as power-aimed training may not impose excessive central fatigue. No significant correlations were found between MDF and velocity loss, indicating that velocity loss alone may be insufficient for comprehensive fatigue monitoring, particularly when measured with lower-precision devices. Therefore, when applying velocity loss as a fatigue indicator in training settings, practitioners may consider combining it with additional markers, such as jump height or RPE, to improve the accuracy of fatigue assessment.

## ACKNOWLEDGEMENTS

ChatGPT 4o was used solely for English proofreading and grammar correction. This did not influence the scientific content or outcomes of the present study. The authors would like to thank all the participants that participated in this study.

### Funding
The authors received no funding for this work.

### Competing Interests
The authors declare there are no competing interests.

### Author Contributions
- Hanye Zhao conceived and designed the experiments, performed the experiments, analyzed the data, prepared figures and/or tables, authored or reviewed drafts of the article, and approved the final draft.
- Takanori Kurokawa performed the experiments, authored or reviewed drafts of the article, and approved the final draft.
- Masayoshi Tajima performed the experiments, authored or reviewed drafts of the article, and approved the final draft.
- Zijian Liu performed the experiments, authored or reviewed drafts of the article, and approved the final draft.
- Junichi Okada conceived and designed the experiments, performed the experiments, analyzed the data, authored or reviewed drafts of the article, and approved the final draft.

### Human Ethics
The following information was supplied relating to ethical approvals (i.e., approving body and any reference numbers):

The experimental protocol was established, according to the ethical guidelines of the Helsinki Declaration and was approved by the Waseda University Human Ethics Committee (Approval No. 2023-112).

### Data Availability
The raw data are available in the Supplemental File.

### Supplemental Information
Supplemental information for this article can be found online at http://dx.doi.org/10.7717/peerj.20426#supplemental-information.

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
