# Peer review of "Understanding perceived exertion in power-aimed resistance training: the relationship between perceived exertion and muscle fatigue"

_PeerJ, doi:10.7717/peerj.20426_

## Round 0.1 · original submission · Major Revisions

· Academic Editor

Major Revisions

The reviewers found the paper of interest but feel the manuscript needs to be substantially revised. Reviewers have concerns that some of the constructs studied are not clearly defined or used correctly. There were also concerns with replicating your analysis. Please address these and the other reviewer comments.

Reviewer 1 ·

Basic reporting

Article is well written and informative.

Experimental design

Need to add a statement explaining the rationale for the 2-second pause between repetitions.

Validity of the findings

Findings were valid and fill a void in the existing literature. This was a strong point of the manuscript.

Additional comments

The rationale and experimental design were well implemented and the conclusions matched the results.

It is unclear why some references were all upper-cased lettering when cited in text.

Although it was specified in the article as to what it represented, BP is a common abbreviation for blood pressure and perhaps BeP or some other abbreviation should be selected.

Reviewer 2 ·

Basic reporting

no comment

Experimental design

no comment

Validity of the findings

no comment

Additional comments

I would like to thank the editor and the authors for the opportunity to review this manuscript. The authors sought to investigate the effects of different exercise volumes on RPE, surface electromyography, and velocity loss. The authors report that there are significant differences in RPE and velocity but not EMG data. While the study is interesting, the authors may need to make substantial changes to the introduction and discussion before it is able to be published.

Abstract:
Line 34-37 “Conclusions: These findings indicate that both athletic performance and RPE are primarily influenced by peripheral fatigue. Practitioners should exercise caution when arranging the training order. However, velocity loss has only limited precision in predicting neuromuscular fatigue.”

Some of these recommendations are not supported by this study. Since the authors only looked at one exercise, it is not possible to make recommendations about exercise order.

Introduction
Lines 51- 53 “Therefore, understanding these subjective discomforts is crucial for developing appropriate resistance training guidelines, achieving optimal athletic performance, and designing long-term periodized training programs.”

This paper only measured RPE, which is different that discomfort. Discomfort is often thought of as afferent feedback, while effort its more related to efferent drive. https://doi.org/10.17338/trainology.6.1_1

Lines 55-61 “55 Subjective displeasure is difficult to capture and varies between individuals; however, in physiological and clinical settings, it can be quantified using the rating of perceived exertion (RPE) scale. RPE is a subjective, perception-based method commonly used to assess exercise intensity (Lagally et al., 2002a; Kilpatrick et al., 2009). Originally developed for evaluating the intensity of rehabilitation and other medically related treatments, the RPE scale has since been widely adopted in physiological and sports science research due to its simplicity and validity in reflecting physiological responses to physical activity.”

This paragraph contradicts itself. The first line states it is difficult to measure, while later in the paragraph it is stated that RPE is can be captured simply and is a valid measurement. That suggest that’s discomfort can be very easy to measure. Secondly, there has been some debate about how accurate RPE is. PMID: 29204323

Line 114-115 “Consequently, the use of RPE in resistance training has been increasing; however, what exactly it measures remains unclear.”

I think this line needs to be rewritten for clarity. I think the authors mean to state that it is not whether RPE is more closely related to central or peripheral fatigue. We have an idea of what it measures, but we may not understand the underlying constructs. RPE measures exertion. The authors are interested in the factors that affect exertion.

Methods
Lines 144-148 “The required sample size was determined using G*Power 3.1 (University of Bonn, Germany) based on an ANOVA model with fixed effects, main effects, and interaction analyses. The input parameters were set as follows: effect size = 0.4, alpha = 0.05, and statistical power = 0.95 (Cruz-Montecinos et al., 2019). The analysis indicated a minimum required sample size of 14 participants. Accordingly, 16 male participants were recruited for this study. “

This is not enough information to replicate the analysis. when I tried to, I was missing numerator DF and the number of groups. Also, this test will not run with only one group.

Discussion
I think the authors need to remove the discussion about displease or show that exertion is unpleasant in everyone. I am not sure that exertion would be considered as a displeasure in all.

Reviewer 3 ·

Basic reporting

Clear, unambiguous, professional English language used throughout - Yes

Intro & background to show context - No (see comments)

Literature well referenced & relevant - Yes

Structure conforms to PeerJ standards, discipline norm, or improved for clarity - Yes

Figures are relevant, high quality, well labelled & described - Yes

Raw data supplied (see PeerJ policy) - Yes

Experimental design

Original primary research within Scope of the journal - Yes

Research question well defined, relevant & meaningful. It is stated how the research fills an identified knowledge gap - No (see comments)

Rigorous investigation performed to a high technical & ethical standard - Yes

Methods described with sufficient detail & information to replicate - Yes

Validity of the findings

All underlying data have been provided; they are robust, statistically sound, &
controlled - Yes

Conclusions are well stated, linked to original research question & limited to supporting results - Partly (see comments)

Additional comments

The majority of my concerns relate to the rationale/justification for the study and the methodology. The introduction is, unfortunately, not an effortless read and I think there could be significant changes made to improve the flow of information. I have also flagged some consistency around use of different terms in the introduction that I believe should be clarified. Please see specific comments below listed by line number:

Introduction, line 42 to 123: The earliest section sets up the narrative that discomfort is bad for performance and adherence to a training regime, but this doesn’t appear to be relevant to the purpose of the study at all (which seems primarily concerned with neuromuscular surrogates of fatigue). Overall, the flow of information could be improved significantly and the section shortened, as it is wordy at times and a challenge to follow. As such, the justification for the study is not inherently clear.

Line 45 to 46: Consider clarifying what ‘fatigue, discomfort and displeasure’ is referring to here in a resistance exercise context. Is it, for example, the immediate discomfort of lifting weights at high volumes (i.e., the physiological response), or is it the delayed soreness associated with muscle damage? I would be hesitant to include the term ‘displeasure’ unless you are able to delineate between the concepts of ‘discomfort’ and ‘displeasure’.

Line 49 to 50: Please consider elaborating on what aspects of athletic performance are impacted to save readers from chasing down the source article.

Line 51: Refining and/or defining the terminology will improve the clarity here. For instance, you indicate that ‘understanding these subjective discomforts is crucial…’, whereas you refer to ‘displeasure’ and ‘unpleasant’ in the sentences prior, and ‘displeasure’ again in line 55.

Line 55 to 57: Citing the meta-analysis by Lea and colleagues would strengthen the introduction (Lea, J.W.D., O’Driscoll, J.M., Hulbert, S. et al. Convergent Validity of Ratings of Perceived Exertion During Resistance Exercise in Healthy Participants: A Systematic Review and Meta-Analysis. Sports Med - Open 8, 2 [2022]). The meta-analyses confirm validity of RPE/RPD (albeit, with a large degree of between-study heterogeneity) in measuring perceived discomfort during resistance exercise and also supports the anchoring technique you have implemented.

Line 75 to 78: There could be several factors contributing to increasing RPE (i.e., not just fatigue [and the type of fatigue]) throughout a resistance exercise set or session – consider elaborating here.

Line 85 to 99: Much of this could be shortened and incorporated with the section above (75 to 83), especially considering it is a very long introduction (~900 words).

Line 90 to 92: Consider rewording this sentence as sEMG is only a surrogate measurement of fatigue, it does not directly measure fatigue.

Line 101 to 102: Clarification recommended. If velocity loss is an indicator of fatigue, does that imply that it is valid? This is later contradicted in lines 108 to 109.

Line 102, 105, 381, 398, 401: Minor comment: Sanchez-Medina citation incorrectly formatted.

Line 109 to 110: Please cite literature to support the ‘unclear’ evidence.

Line 114 to 123: What is your hypothesis and why is validating the measures important for practitioners?

Line 119 to 123: Regarding aim 1, the concern I have is that you are hoping to draw conclusions from a small number of related surrogate measurements. For example, you are aiming to determine whether RPE (i.e., exertion, displeasure, discomfort [as described above]) is related to neuromuscular activity via sEMG is related to central or peripheral fatigue.

Line 178 to 179: How exactly were the participants asked for their RPE? This is important because concepts listed here (i.e., exertion, fatigue, and discomfort) do have a degree of independence from one another. For example, performing a 1RM may require maximal exertion, but would be less likely to stimulate the sensations of discomfort or physiological fatigue associated with higher volume sets. Likewise, mechanical strain associated with acute stretching can cause discomfort (Laur DJ, Anderson T, Geddes G, Crandall A, Pincivero DM. The effects of acute stretching on hamstring muscle fatigue and perceived exertion. J Sports Sci. 2003;21(3):163–70.) but would not be associated with fatigue.

Line 160 to 195: To confirm, is it correct that participants first performed a 1RM assessment, then rested for 5 minutes prior to performing the ‘as many reps as possible’ (AMRAP) trial at 65% of 1RM? If so, how confident are you that the 1RM assessments prior had no influence on their performance on the AMRAP set? Is there a reason why it was performed this way, particularly given that this study has fatigue-related outcomes? This is particularly important because the sets based off AMRAP performance (i.e., L, M and H trials) were performed on a separate day.

Line 219 to 220: I am assuming participants reported RPE for each repetition – please clarify.

Line 224: Again, I have concerns with the fact that the three trials were performed within the same session. Is 5 minutes of rest adequate? Is it an arbitrary amount of time, or evidence-based? I understand you have counterbalanced the order, but the issue only occurs when participants are performed H to M to L, or H to L to M order (i.e., the high-volume set would likely negatively impact all other sets if performed first, but not the other way around).

Line 255 to 267: What is the rationale for using this method of assessing movement velocity over established and validated methods (e.g., an LPT)?

Line 284: For the validity analysis, was an a priori level of acceptable validity established?

Line 328 to 329: Are you sure you are able to determine this from the study that was conducted? You have determined the relationship between RPE and one or two contributors (i.e., neuromuscular fatigue inferred from changes in sEMG measures or velocity loss). The weighting of its contribution relative to other contributing factors remains unknown.

Line 377: Again, sEMG measures surface electrical activity but not neuromuscular fatigue directly.

Line 404 to 406: How precise does a device have to be? For example, a GymAware operates at 50Hz and has been validated (Weakley, J., Morrison, M., García-Ramos, A., Johnston, R., James, L., & Cole, M. H. (2021). The Validity and Reliability of Commercially Available Resistance Training Monitoring Devices: A Systematic Review. Sports medicine (Auckland, N.Z.), 51(3), 443–502). Is the difference in precision between your method and established devices meaningful?

Figure 1: Consider representing data using a scatter (Weissgerber, T. L., Milic, N. M., Winham, S. J., & Garovic, V. D. (2015). Beyond bar and line graphs: time for a new data presentation paradigm. PLoS biology, 13(4), e1002128).

Reviewer 4 ·

Basic reporting

The manuscript addresses an interesting and relevant topic in the literature, with potential to contribute to the understanding of subjective responses during exercise. However, the narrative is unclear and lacks a coherent linear structure, particularly in the introduction. Key terms such as resistance training, velocity-based training, and power training are used interchangeably without a clear definition or distinction, leading to conceptual confusion.
Furthermore, the authors appear to conflate perceived exertion (RPE) with pleasure/displeasure. RPE is not an affective measure but an effort scale; thus, its role in this context needs to be better defined. One paragraph in the introduction focuses on RPE, while another discusses EMG, but these are not presented synergistically. The link between physiological measures and the subjective displeasure construct is not convincingly established.

I recommend a thorough review of the conceptual differences between resistance, velocity-based, and power training, and the distinction between affective responses (pleasure/displeasure) and perceived exertion.

Experimental design

In general, the method is good, and I have only appointments in the EMG section. Did the authors report summing EMG activity from the pectoralis, triceps, and deltoid muscles? This approach is unclear. How were the signals from different muscles combined? Considering that fatigability varies between muscles, this summation could introduce bias and compromise the validity of the results.

Validity of the findings

no comment

Additional comments

The manuscript needs substantial conceptual clarification before it can be considered for publication. While the topic is relevant, the confusion between constructs, lack of clear operational definitions, and methodological ambiguities limit its contribution to the field. A deep revision of the text, particularly in the introduction, is recommended to ensure terminological accuracy and alignment between aims, design, and interpretation.

---

## Round 0.2 · Minor Revisions

· Academic Editor

Minor Revisions

Please address the few remaining reviewer comments. In addition, given the repeated measures of your data, please make sure the dots are connected in all of your figures. This is essential otherwise your individual plots are not informative.

Reviewer 1 ·

Basic reporting

This was excellent throughout the article.

Experimental design

The only question I have is: was form/technique monitored and standardized? This was not clear in the methods section. Any alterations in range of motion could affect results.

Validity of the findings

Important investigation to fill a void in the literature.

Additional comments

Overall, a well-done study.

Reviewer 2 ·

Basic reporting

none

Experimental design

Thank you for opportunity to re-review this manuscript. While the authors address most of my comments, I am not convinced their power analysis is correct. I think it may be more appropriate to use an ANOVA with repeated measures with within factors. I would suggest the authors address this in the limitations.

Validity of the findings

none

Reviewer 3 ·

Basic reporting

No comment

Experimental design

No comment

Validity of the findings

No comment

Additional comments

The authors have done a satisfactory job at addressing the concerns of the reviewers. I only have one additional comment for consideration: The first line of the abstract reads "Perceived exertion is one of the unexpected outcomes of power-aimed resistance training". It's an interesting choice (one that I find bizarre) to suggest that exertion is an unexpected outcome of lifting something heavy.

Reviewer 4 ·

Basic reporting

The authors sufficiently clarify my comments.

Experimental design

No comments

Validity of the findings

The authors sufficiently clarify my comments.

Additional comments

The authors sufficiently clarify my comments.

---

## Round 0.3 · Minor Revisions

· Academic Editor

Minor Revisions

I made this comment last time but it does not appear to have been addressed. given the repeated measures of your data, please make sure the dots are connected in all of your figures. This is essential otherwise your individual plots are not informative.

Reviewer 2 ·

Basic reporting

no comment

Experimental design

no comment

Validity of the findings

no comment

Additional comments

The authors have fully addressed my comments.

---

## Round 0.4 · accepted · Accept

· Academic Editor

Accept

The reviewers have addressed reviewer comments. The paper is ready for publication.